To be submitted: Geoscientific Model Development

Development of a Model Framework for Terrestrial Carbon Flux Prediction: the Regional Carbon and Climate Analytics Tool (RCCAT) Applied to Non-tidal Wetlands

Ashley Brereton<sup>1,2</sup>, Zelalem A Mekonnen<sup>1</sup>, Bhavna Arora<sup>1</sup>, William J. Riley<sup>1</sup>, Kunxiaojia Yuan<sup>1</sup>, Yi Xu<sup>2</sup>, Yu Zhang<sup>3</sup>, Qing Zhu<sup>1</sup>, Tyler L. Anthony<sup>4</sup>, Adina Paytan<sup>2</sup>

Earth and Environmental Sciences Area, Lawrence Berkeley National Laboratory, Berkeley, CA, USA<sup>1</sup>

Department of Earth and Planetary Sciences, University of California at Santa Cruz, Santa Cruz, CA, USA<sup>2</sup>

Earth and Environmental Sciences Division, Los Alamos National Laboratory, Los Alamos, New Mexico, USA<sup>3</sup>

California Department of Water Resources, Sacramento, CA, USA<sup>4</sup>

#### Abstract

Wetlands play a pivotal role in carbon sequestration but emit methane (CH<sub>4</sub>), creating uncertainty in their net climate impact. Although process-based models offer mechanistic insights into wetland dynamics, they require extensive site-specific parameterisation (e.g. soil carbon profiles, pore-water chemistry, vegetation-specific model parameters), as well as highresolution hydrological and meteorological inputs that are often difficult to obtain outside of wellinstrumented research sites, which makes regional upscaling challenging. In contrast, datadriven models provide a scalable alternative by leveraging available datasets to identify patterns and relationships, making them more adaptable for large-scale applications. However, their performance can vary significantly depending on the quality and representativeness of the data, as well as the model design, which raises questions about their reliability and generalizability in complex wetland systems. To address these issues, we present a data-driven framework for upscaling wetland CO<sub>2</sub> and CH<sub>4</sub> emissions, across a range of machine learning models that vary in complexity, validated against an extensive observational dataset from the Sacramento-San Joaquin Delta. We show that artificial intelligence (AI) approaches, including Random Forests, gradient boosting methods (XGBoost, LightGBM), Support Vector Machines (SVM) and Recurrent Neural Networks (GRU, LSTM), outperform linear regression models, with RNNs standing out, achieving an R<sup>2</sup> of 0.73 for daily CO<sub>2</sub> flux predictions compared to 0.64 for linear regression, and an R<sup>2</sup> of 0.53

for CH<sub>4</sub> flux predictions compared to 0.47 for linear regression. Interestingly, linear regression performed better than random forest for methane flux, which highlights the necessity for

comparison. Despite that, interannual variability is less well captured, with annual mean absolute error of 176 gC m<sup>-2</sup> yr<sup>-1</sup> for CO<sub>2</sub> fluxes and 9 gC-CH<sub>4</sub> m<sup>-2</sup> yr<sup>-1</sup> for CH<sub>4</sub> fluxes. By integrating vertically-resolved atmospheric, subsurface, and spectral reflectance information from readily available sources, the model identifies key drivers of wetland CO<sub>2</sub> and CH<sub>4</sub> emissions and enables regional upscaling. These findings demonstrate the potential of AI methods for upscaling, providing practical tools for wetland management and restoration planning to support climate mitigation efforts.

### 1. Introduction

Wetlands provide a wide array of ecological, economic, and environmental benefits (Costanza et al., 2014). They play a crucial role in biodiversity conservation, water purification, flood control, and climate regulation (Grande et al., 2023; Sharma and Singh, 2021). Significant attention has been recently given to wetland restoration due to their ability to sequester carbon from the atmosphere (Lolu et al., 2020; Upadhyay et al., 2020). These ecosystems are highly effective at storing carbon in their soils because the anaerobic conditions in waterlogged soils suppress organic matter decomposition, allowing carbon to accumulate over time (Mitsch and Gosselink, 2015a). However, wetlands can also be significant sources of CH<sub>4</sub>, a potent greenhouse gas (Brix et al., 2001), leading to potentially net positive effects of wetlands on climate warming. The most accurate way to determine the carbon balance in natural ecosystems is through direct and continuous measurements of carbon and GHG sources and sinks (Baldocchi et al., 2001). This involves monitoring carbon dynamics using techniques such as eddy covariance (EC) towers (Aubinet et al., 2012), soil carbon stock assessments (Harrison et al., 2011), and lateral carbon transport measurements (Ciais et al., 2008). However, these measurements are time-consuming to carry out, costly, and require specialized instruments and expertise, limiting their application to a few representative sites globally (Hill et al., 2017; Kumar et al., 2017). The Ameriflux network offers roughly 500 EC sites comprising about 3600 site years of data, monitoring carbon fluxes across various ecosystems such as forests, grasslands, and wetlands (Pastorello et al., 2020). Eddy-covariance site footprints range in scale and are typically determined by the sensor height and atmospheric turbulence (Chu et al., 2021). Data from these Ameriflux sites could potentially be upscaled and used for estimating fluxes from non-monitored sites to obtain regional assessments of carbon balance for various ecosystem types, including wetlands.

In this study, we focus on nontidal wetlands due to the presence of a cluster of EC towers in a small region located in the Sacramento-San Joaquin Delta, including three sites, each with over a decade of continuous data. Reported sequestration rates in wetlands vary widely, influenced by factors such as climate, vegetation, and management. For instance, reported sequestration rates range from as low as 26 gC m<sup>-2</sup> yr<sup>-1</sup> in boreal rain-fed bogs (Villa and Bernal, 2018) to as high as 797 gC m<sup>-2</sup> yr<sup>-1</sup> in constructed wetlands with emergent *Phragmites* in the Netherlands (de Klein and van der Werf, 2014). Similarly, temperate wetlands in central Ohio exhibit a wide range of carbon sequestration rates depending on vegetation: forested depressional wetlands dominated by *Quercus palustris* sequester up to 473 gC m<sup>-2</sup> yr<sup>-1</sup>, while marshes dominated by

*Typha* sequester around 210 gC m<sup>-2</sup> yr<sup>-1</sup> (Bernal and Mitsch, 2012). In Victoria, Australia, freshwater marshes show varying sequestration rates from 91 gC m<sup>-2</sup> yr<sup>-1</sup> in shallow marshes to 230 gC m<sup>-2</sup> yr<sup>-1</sup> in permanent open freshwater wetlands (Carnell et al., 2018). More relevant to this study, in the San Francisco Bay-Delta region, nontidal managed wetlands dominated by Schoenoplectus and Typha species sequester carbon at rates of approximately 355  $\pm$  249 gC-CO2 m<sup>-2</sup> yr<sup>-1</sup>. This estimate is based on direct calculations using Ameriflux data from sites with over a decade of observations (US-Myb, US-Tw1, and US-Tw4). For this calculation we used full-year annual averages and their corresponding standard deviation to the annual mean, to highlight the significant inter-annual variability, with the standard deviation close to the mean. The unit reported for these Delta sites is in gC-CO<sub>2</sub> m<sup>-2</sup> yr<sup>-1</sup>, as the EC tower directly detects CO<sub>2</sub> exchange, which is convenient for GHG assessment purposes. It is worth noting that, at these sites, some years were a net CO<sub>2</sub> source, due to site-specific disturbances such as caterpillar infestations, drought, or when vegetation cover was fully established (Anderson et al., 2018; Knox et al., 2017; Rey-Sanchez et al., 2021) . See table S1 for more detailed information and references therein.

Although CO<sub>2</sub> balance (photosynthesis minus community respiration) is an important component of carbon sequestration, in many wetland systems sequestration benefits are counterbalanced by CH<sub>4</sub> emissions, a potent greenhouse gas, with a warming potential 27 times higher than CO<sub>2</sub> (Lee et al., 2023) that can often offset climate mitigation efforts. CH<sub>4</sub> emission rates also vary substantially over time and across wetlands, from as low as 0.23 qC-CH<sub>4</sub> m<sup>-2</sup> yr<sup>-1</sup> in saltwater zones of estuarine environments (Abril and Iversen, 2002) to as high as 270 gC-CH<sub>4</sub> m<sup>-2</sup> yr<sup>-1</sup> in certain freshwater wetlands (Knox et al., 2021). For example, restored freshwater wetlands in Maryland dominated by grasses and sedges emit around 142 gC-CH<sub>4</sub> m<sup>-2</sup> yr<sup>-1</sup> (Stewart et al., 2024). Tropical wetlands in Costa Rica exhibit some of the highest emissions, with isolated and floodplain wetlands releasing between 220 and 263 gC-CH<sub>4</sub> m<sup>-2</sup> yr<sup>-1</sup> (Mitsch et al., 2013). The San Francisco Bay-Delta wetlands that have high carbon sequestration rates also release CH<sub>4</sub> at rates of 35 ± 13 gC-CH<sub>4</sub> m<sup>-2</sup> yr<sup>-1</sup> (direct measurements from the eddy covariance tower data (Arias-Ortiz et al., 2021)). See table S2 for further information and reference therein. This dual role of wetlands in both sequestering carbon and emitting CH₄ reveals the complex effect they have on the global greenhouse gas balance. Therefore, integrating CO2 and CH4 emissions is critical to assess the net climate benefits of wetland conservation and restoration initiatives.

To evaluate how wetlands contribute to the atmospheric radiation budget at larger scales, it is essential to quantify both GHG emissions and carbon sequestration, especially at sites where direct measurements are unavailable (Moomaw et al., 2018). Upscaling models serve this purpose by allowing estimation of sequestration and emission rates across larger spatial scales than those covered by the original data sources (Villa and Bernal, 2018) which provide GHG accounting and net climate benefit assessments for specific wetland sites (Nahlik and Fennessy, 2016). Moreover, it aids in targeting wetland restoration efforts that aim to optimize sequestration by identifying locations with the greatest potential for net carbon uptake.

Process-based models have traditionally been used to estimate sequestration and emissions (Mack et al., 2023; Zhang et al., 2002). Models such as DNDC (Li, 1996), DayCent (Parton et

al., 1998), and *Ecosys* (Grant et al., 2017) have been applied to simulate biogeochemical processes in terrestrial ecosystems, including modeling CH<sub>4</sub> emissions, carbon balances, and soil carbon and nitrogen cycling (Grant and Roulet, 2002; Weiler et al., 2018; Zhang et al., 2002). While these models can elucidate the processes that play a role in carbon dynamics, they require extensive mechanistic parameterization to accurately represent the interactions in various ecosystems(Pastorello et al., 2020; Yin et al., 2023). This approach often necessitates site-specific information and data collection, making implementation over vast areas challenging (Saunois et al., 2024; Xu and Trugman, 2021). The extensive data needs associated with these process-rich models showcase the need for alternative approaches that can effectively upscale wetland emissions without such intensive resource demands.

Artificial Intelligence (AI) methods, such as machine learning and deep learning, have been widely applied in ecological modeling in recent years, alongside long-term, large-scale data collection efforts (Perry et al., 2022). Recent deep learning applications have demonstrated success in capturing the complex dynamics of carbon and methane fluxes in these systems (Ouyang et al., 2023; Yuan et al., 2022, 2024; Zou et al., 2024). The availability of open-source modeling platforms like TensorFlow and PyTorch has made advanced computational techniques, such as neural networks, more accessible, enabling the rapid development and deployment of a range of specialized modeling tasks (Xu et al., 2021). Despite several recent studies demonstrating the potential of machine learning for large-scale carbon cycling in wetland ecosystems, this remains a relatively young field. Moreover, carbon dynamics in wetland ecosystems are temporally variable and inherently nonlinear, making them particularly well-suited for testing machine learning approaches (Arora et al., 2019, 2022). We therefore emphasize the importance of evaluating and comparing various approaches within this domain and their potential for large-scale assessment.

A pervasive challenge in model development is the ability to balance complexity with generalizability. While more complex models can capture nonlinear relationships, they also increase the risk of overfitting, where the model performs well in the testing, but poorly on new conditions (Hastie, 2009; Tashman, 2000). Furthermore, it is also important to use a robust validation framework. For the application of upscaling, it is important that the model is able to extrapolate spatially. For this purpose, a leave-one-site-out (LOSO) validation approach is typically carried out, whereby the models are trained on data that excludes a single site, with the excluded site data saved for model testing (Bodesheim et al., 2018; Tramontana et al., 2016). It is also important to avoid data leakage, where information from the training set inadvertently appears in the testing set (Kaufman et al., 2012), a risk posed when splitting temporally adjacent data points that are close in value, potentially inflating performance statistics (Bergmeir and Benítez, 2012; Kaufman et al., 2012). For example, daily rates of change relative to a system where seasonal dynamics dominate, such as emissions of CH<sub>4</sub> emissions in vegetated wetlands (Knox et al., 2021).

In this study, we introduce a model framework for coastal nontidal wetland CO<sub>2</sub> and CH<sub>4</sub> emissions using several 'off-the-shelf' models. These models are trained and validated against observational data, and results are compared to find the most predictive model. The top performing model is then used to upscale carbon sequestration and CH<sub>4</sub> emissions in nontidal

wetlands at regional scale. The San Francisco Bay-Delta serves as the area of interest, due to its network of EC towers that have been operating for a relatively long time and relevance to future wetland restoration efforts. We employ a suite of models, ranging widely in complexity: (1) linear regression; (2) Random Forests (Breiman, 2001), an ensemble method that constructs multiple decision trees to reduce overfitting; (3) gradient boosting techniques such as LightGBM (Ke et al., 2017) and XGBoost (Chen and Guestrin, 2016), which are scalable tree boosting systems able to handle complex nonlinear relationships and variable interactions; (4) Support Vector Machines (SVM) (Cortes, 1995), a kernel-based technique that can approximate nonlinear boundaries between data points and (5) the Recurrent Neural Network (RNN) such as the Long Short-Term Memory (LSTM) neural network (Hochreiter, 1997), an advanced model designed to process sequential data and capture non-linear interactions over long-term dependencies. We also test a model with similar but simpler architecture, the Gated Recurrent Unit (GRU) (Chung et al., 2014), which uses fewer parameters. Linear regressions serve as a baseline to assess the applicability of the more sophisticated methods. Random Forests have been used to upscale northern wetland methane emissions (Peltola et al., 2019), gradient boosting methods have demonstrated success in ecological modeling (Ding, 2024; Räsänen et al., 2021; Zou et al., 2024), and LSTM neural networks have been successfully applied to model CO<sub>2</sub> and CH<sub>4</sub> fluxes in ecosystems (Yuan et al., 2022, 2024; Zou et al., 2024). Our proposed framework is designed to provide transparency, easy determination of model practicality and applicability, and contextualisation to model performances by comparing to a baseline model (i.e. linear regression).

## 2. Methods

Our ultimate aim is to establish a robust modeling framework for estimating wetland carbon fluxes in sites that are not monitored. To achieve this, we compare a range of models, from simple linear regression to advanced recurrent machine learning neural networks. Since the goal is to predict unseen sites, we emphasize cross-site predictability by validating and testing the models at sites not included in training. Doing so ensures predictions are applicable beyond the training sites and addresses challenges often associated with model generalizability (Meyer and Pebesma, 2022). This strategy serves several purposes:

- 1. **Performance Contextualization**: Starting with the simplest type of model provides a baseline for performance and helps evaluate the advantage (or lack thereof) for using more complex models.
- Practicality and Transparency: Advanced models may offer better performance but
  often require significant effort to set up and may lack interpretability. By comparing
  models of varying complexity using the same input data, we assess whether the added
  complexity is justified.
- 3. **Feature Evaluation**: Training with different combinations of relevant features helps us to understand which features are dominating control, and the limitations of the data in terms of predictive capacity.

# 2.1 Model targets

The model targets two key variables: CO<sub>2</sub> (**FCO2**) and CH<sub>4</sub> (**FCH4**) surface emissions. Both variables follow a sign convention where positive values indicate emissions to the atmosphere (source) and negative values indicate sequestration (sink). Both variables are available at half-hourly resolution through the Ameriflux database.

The models we developed all operate on a daily time scale, requiring target variables to be aggregated to the daily time scale. This approach assumes that sub-daily variations have a negligible non-linear contribution to longer time scales, an assumption supported by the dominant seasonal signal typically observed in flux data from these systems (Knox et al., 2021).

These target variables could then be used to calculate annual NECB (Net Ecosystem Carbon Balance; gC m<sup>-2</sup> yr<sup>-1</sup>) and annual wetland net atmospheric radiative effect (FCO<sub>2</sub>e (CO<sub>2</sub>-equivalent flux) gCO<sub>2</sub>e m<sup>-2</sup> yr<sup>-1</sup>). The global warming potential (GWP) of non-fossil CH<sub>4</sub> is 27.2 as per the latest IPCC assessment(Lee et al., 2023). For this study, we neglect contributions of lateral fluxes due to data limitations, and that lateral transport at these sites is assumed to be negligible due to the limited outflow from the wetlands (Miller et al., 2008). FCO2e is defined as annually averaged CO<sub>2</sub> and CH<sub>4</sub> emissions, adjusted for the global warming potential (GWP) of each gas. A positive FCO<sub>2</sub>e indicates that the ecosystem is contributing positively to atmospheric warming, and vice versa. Here we consider CO<sub>2</sub> and CH<sub>4</sub> emissions but neglect contributions from N<sub>2</sub>O due to data limitations and because N<sub>2</sub>O emissions are considered negligible in Delta wetlands (Windham-Myers et al., 2018).

# 2.2 Region of interest

The Sacramento-San Joaquin Delta was selected for this study due to its high density of EC towers and extensive long-term data. We selected sites for model training and validation where data was collected for at least a decade to capture interannual variability. Hence three restored wetland sites, US-Myb (Matthes et al., 2016), US-Tw1 (Valach et al., 2016), and US-Tw4 (Eichelmann et al., 2016) are selected in this study. While data from two other sites (i.e., US-Sne and US-Tw5) are available, the lack of sufficient temporal coverage and, in the case of US-Sne, not fully established vegetation cover, makes them less representative of a stable ecosystem. Focusing on sites with over a decade of continuous data allows for capturing long-term dynamics more effectively and provides sufficient time for the wetlands to reach a stable state. The dataset encompasses 35 full site-years of observations across the three sites within the Delta (Novick et al., 2018) (Table 2, Figure 1), with detailed mapping data sourced from the Ecoatlas Database (Workgroup, 2019) which provides land use and vegetation surveys across wetlands in California.

Table 2: Model training sites

| Site   | Site Name                            | Water     | Salinity | Years of    | Start |
|--------|--------------------------------------|-----------|----------|-------------|-------|
| Code   |                                      | Туре      |          | Data (Full) | Date  |
| US-Myb | Mayberry Wetland                     | Non-Tidal | Fresh    | 13          | 2010  |
| US-Tw1 | Twitchell Wetland West Pond          | Non-Tidal | Fresh    | 12          | 2011  |
| US-Tw4 | Twitchell Island East End<br>Wetland | Non-Tidal | Fresh    | 10          | 2013  |

The sites are dominated by Tules (*Schoenoplectus*), Cattails (*Typha*), and invasive species such as *Phragmites*, which are perennial emergent plants well suited to wetland environments (López et al., 2016). The Delta itself is host to the largest estuarine system on the US Pacific coast, spanning approximately 3,000 km², and contains a diverse network of wetland systems. Historically, much of the area was drained and converted for agriculture (Laćan and Resh, 2016; Lund et al., 2010), but recent restoration efforts have reclaimed select portions of the landscape for environmental benefits.

Figure 1: Map of the Sacramento-San Joaquin Delta's wetland system. The Eddy-covariance tower site locations outlined in Table 2 are shown in the red and purple boxes. Satellite image: © Google Earth, accessed 2025.

#### 2.3 Model features

The application of this work focuses on upscaling carbon fluxes from similar wetlands at a regional scale. To achieve this, we aim to predict fluxes at unmonitored sites using widely available data that are expected to be key drivers of FCO2 and FCH4. Since site-level measurements from EC towers are not available at a larger spatial scale, we focus on ecosystem drivers that can be accessed across broader spatial extents.

The models utilize a comprehensive set of features from two readily accessible datasets: (i) the Western Land Data Assimilation System (WLDAS) (Erlingis et al., 2021) and (ii) Landsat surface-reflectance products (Landsat, 2020). A list of features can be found in Supplementary Table S3. Initially surface reflectance products were derived from MODIS (Justice et al., 2002), but we found better model performance with Landsat features. WLDAS provides hydrological and meteorological data at 1 km spatial and daily temporal resolution; we bilinearly interpolate these fields to each tower coordinate (no additional smoothing). Landsat offers 30 m pixels at a nominal 16-day revisit, although temporal resolution increases with time as more satellites are added; we average a 3 × 3 pixel window centred on the tower, linearly interpolate the series to daily resolution, and apply a centred 17-day running mean to improve data continuity.

#### 2.4 Model suite

To evaluate ML model performance in calculating FCO2 and FCH4, we implemented a suite of seven models ranging from simple linear methods to more complex neural networks. These models have been used in various ecosystems to study fluxes and collectively represent a broad spectrum of methodological complexity. Table 3 summarizes the core characteristics and advantages of each approach.

Table 3: An overview of the models that are applied to wetland fluxes

| Model Name | Category | Description | Key Strengths |
|------------|----------|-------------|---------------|

| Linear Regression                                 | Regression                    | Fits a linear relationship<br>between predictors and<br>fluxes     | Simple baseline,<br>easily<br>interpretable<br>(Breiman, 2001)      |
|---------------------------------------------------|-------------------------------|--------------------------------------------------------------------|---------------------------------------------------------------------|
| Random Forest<br>(Breiman, 2001)                  | Ensemble of<br>Decision Trees | Aggregates multiple decision trees to enhance prediction stability | Robust to<br>nonlinearity,<br>reduces overfitting<br>(Cortes, 1995) |
| Support Vector<br>Machine (Cortes,<br>1995) (SVM) | Kernel-Based<br>Method        | Uses flexible kernels to find optimal separating hyperplanes       | Effective in high dimensions, adaptable kernels (Ke et al., 2017)   |
| LightGBM (Ke et al., 2017)                        | Gradient<br>Boosting          | Employs iterative boosting with efficient tree growth              | Fast, memory-<br>efficient, handles<br>large datasets               |
| XGBoost (Chen and<br>Guestrin, 2016)              | Gradient<br>Boosting          | Improves boosting with regularization and efficient computations   | Manages outliers,<br>handles sparse<br>data well                    |
| LSTM Neural<br>Network<br>(Hochreiter, 1997)      | Recurrent<br>Neural Network   | Captures temporal dependencies in sequential data inputs           | Ideal for time-<br>series, learns<br>long-term patterns             |
| GRU Neural<br>Network (Chung et<br>al., 2014)     | Recurrent<br>Neural Network   | Similar to LSTM but streamlined with fewer parameters              | Efficient temporal<br>modeling, lower<br>complexity                 |

These models act to demonstrate a spectrum of model complexity and how that can be leveraged to improve flux prediction.

After performing simple grid searches we found that all models were largely insensitive to hyperparameter tuning, so we kept almost everything at the package defaults with some minor exceptions. Model hyper-parameter choices can be found in Brereton (2025).

## 2.5 Validation framework

To evaluate the models' ability to generalize across sites, we employed a Leave-One-Site-Out (LOSO) cross-validation strategy. In LOSO, we train the models on data from all but one site, and test the models on the excluded site. This approach is repeated for each site in the dataset

and then aggregated, ensuring that there are no spatio-temporal connections between the training and testing data. While few models are immune to overfitting, this approach minimizes the risk of doing so.

An integral part of our modeling approach is the strategic selection of input features to optimize the model's performance. We perform this selection by first selecting features that are expected to be important, guided by mechanistic considerations of wetland processes gained from fieldwork and insights from mechanistic models (Table S3). Since the total number of possible feature combinations is too large for an exhaustive search, we adopt a feed-forward selection (**FFS**) strategy. This method begins with a single feature and iteratively adds features that most improves the model's performance based on a chosen statistic. At each step, we evaluate the model's performance with each potential new feature and select the one that provides the greatest improvement. This process continues until adding additional features no longer significantly enhances the model's performance. By using this approach, we efficiently identify the most influential predictors without the computational burden of testing all possible combinations.

#### 2.6. Validation

As suggested above, each model was trained using data from two wetland sites and then validated on the third. Although the number of sites was limited, each site offered over a decade of observations accumulated to a daily time step, ensuring exposure to a range of environmental conditions representative of the wetland type and regional climate. For each excluded site, the model's predictions were compared against measured FCO2 and FCH4 and we calculated R², Pearson's r, and RMSE for that site. We then pooled all held-out predictions from the three sites into one combined set and recomputed R² (as well as r and RMSE) on the full array to give an overall cross-validation score. This process was paired with the FFS method optimized to maximize R².

After selecting LSTM as the model of choice, it was retrained using all available data from the three sites for upscaling. The Sacramento-San Joaquin Delta contains roughly 700km2 of wetland area, including tidal and nontidal regions. The upscaling domain encompasses approximately 25 km² of nontidal wetlands in the region, dominated by vegetation types relevant to the training sites, specifically Tules, Cattails, and Phragmites. The assumption is that the training sites used in this study are representative of the broader conditions in the Delta, but we acknowledge that local variability in carbon dynamics, such as those caused by microclimates prevalent in the area, may not be fully captured during the ML model training. Improvements to the model might be achieved if additional site data covering a wider range of environmental conditions were incorporated. The feature data used to optimize the model were spatially interpolated onto the regional model grid and the model applied to yield flux estimations. Although relatively modest in spatial extent, these wetlands are of particular interest given their role in carbon sequestration and potential climate mitigation and as targets for conservation and restoration.

### 3. Results

#### 3.1 Model Validation

We tested six modeling techniques of varying complexities (Table 3). Model performance scores for daily predictions are shown in Figure 2, demonstrating that nearly all machine learning models outperformed the linear regression baseline (R² = 0.64 for FCO2 and R² = 0.47 for FCH4). For FCO2, LSTM and GRU achieved the highest R² values (0.73 and 0.71, respectively), outperforming other methods. A similar result was found for FCH4, with LSTM and GRU both scoring R² of 0.53. These results suggest that deep learning models can provide tangible benefits over linear regression methods for upscaling flux predictions. The LSTM model was selected for upscaling in this study as it scored highest consistently, though other ML models scored comparably, so we do not assert it as definitively the best model.

The feature selection process had access to 26 environmental features from WLDAS and 7 features derived from LANDSAT spectral bands(see table S3 for full details). These variables encompass a wide range of atmospheric, soil, and vegetation characteristics, such as precipitation, temperature, soil moisture, and spectral indices, key environmental drivers known to influence carbon and methane flux dynamics (Mitsch and Gosselink, 2015b).

The feature selection routine converged on variables that map directly onto the three main controls of wetland carbon cycling - vegetation productivity, surface energy-water balance, and microbial temperature sensitivity, see Table 4. For FCO2, the features selected were the Soil-Adjusted Vegetation Index (SAVI) and the upwards sensible heat flux, which are proxies for gross primary production and the surface energy water balance (Anderson et al., 2016; Huete, 1988). For FCH4, the features selected were canopy temperature, soil temperature and Greenness Difference Vegetation Index (GNDVI), which are proxies for short-term thermal forcing and vegetation water status, the anaerobic root-zone temperature that governs methanogenesis, and the supply of photosynthetically derived substrates for microbes, respectively (Bubier et al., 1993; Knox et al., 2021; Whiting and Chanton, 1993; Yvon-Durocher et al., 2014).

Table 4: Feed-forward feature selection process.

| Target S | Step | Chosen Feature | R² | RMSE | r |
|----------|------|----------------|----|------|---|
|----------|------|----------------|----|------|---|

| FCO2 | 1 | Soil-adjusted<br>vegetation index<br>(SAVI)  | 0.59 | 1.79  | 0.78 |
|------|---|----------------------------------------------|------|-------|------|
| FCO2 | 2 | (Upwards)<br>sensible heat flux              | 0.73 | 1.46  | 0.86 |
| FCH4 | 1 | Canopy<br>Temperature                        | 0.48 | 0.054 | 0.70 |
| FCH4 | 2 | Soil temperature (10-40 cm)                  | 0.52 | 0.052 | 0.73 |
| FCH4 | 3 | Normalized Difference Greenness Index (NDGI) | 0.53 | 0.051 | 0.74 |

Figure 3 shows both FCO2 and FCH4 results, including time series and scatter plots comparing predictions to observations. Overall, the predicted values track the observations reasonably well. For FCO2, predictions tended to regress toward the mean, underestimating peak emissions at local maxima and overestimating at local minima, although reasonable interannual variability was observed. The ML models also displayed less interannual variability than the observations, common in machine learning approaches (Ouyang et al., 2023). For wetlands, this is likely due to limited subsurface process information included in the machine learning models. Still, the scatter plot shows strong performance for FCO2 (r = 0.86,  $R^2 = 0.73$ , RMSE = 1.46 gC-CO<sub>2</sub>  $m^{-2}$  day<sup>-1</sup>), despite a noticeable spread around the 1:1 line.

FCH4 predictions exhibited similar behavior, with low interannual variability than the observations. At the US-Myb site, for example, observed FCH4 were initially high (aside from the first year, when vegetation cover had yet to be fully established) but declined over time, stabilizing at lower values. The ML models captured this shift to some extent, predicting higher fluxes early in the time series and then modulating to lower levels later on. However, predictions did not fully replicate the magnitude of the observed downward annual trend, introducing bias into the scatter plots at higher and lower extreme values. This phenomenon is known as regression to the mean, observed in similar machine learning studies (Ouyang et al., 2023). Consequently, the FCH4 model performance was weaker than the FCO2 model (R² = 0.53, r = 0.74, RMSE = 0.05 g C-CH4 m² day⁻¹), indicating that the processes controlling FCH4 in younger wetlands like US-Myb may require more detailed subsurface information (such as soil organic C, oxygen, or redox information) to be accurately modeled. Restored Delta wetlands are often net GHG sources for 1-3 years after flooding, before vegetation is fully established. Eddycovariance measurements show positive NEE of +201 ± 101 g C-CO₂ m⁻² yr⁻¹ and elevated

CH<sub>4</sub> emissions in the initial period, switching to sinks of between -400 to -700 g C-CO<sub>2</sub> m<sup>-2</sup> yr<sup>-1</sup> thereafter (Hemes et al., 2019). A larger synthesis found that this can persist decades in nontidal marshes because CH<sub>4</sub> radiative forcing outweighs CO<sub>2</sub> burial (Arias-Ortiz et al., 2021). Similar contrasts between 2 and 15-year-old wetlands (Knox et al., 2015).

The annual bar plots presented in Figure 4 highlight the model's difficulty in capturing the interannual variability of carbon fluxes across the study sites. While the average FCO2 and FCH4 predictions are generally aligned with observed average values with small overall mean bias, the model struggles to reproduce the observed year-to-year variability. Although direct subsurface measurements are available at certain sites, at the regional scale their limited spatial and temporal coverage currently limits integration into models designed for regional upscaling over inter-annual timescale. For example, while spatial maps of wetland soil organic carbon exist (Uhran et al., 2022), using only three sites for training purposes would provide just three corresponding data points, limiting model training. The LOSO validation approach revealed that deep learning models, particularly LSTM and GRU, consistently outperformed traditional linear regression and other machine learning methods for both FCO2 and FCH4 predictions. While nonlinear models demonstrated clear advantages, the magnitude of improvement was relatively modest, reflecting the inherent challenges of capturing site-specific inter-annual dynamics of wetland emissions. To improve model performance, additional techniques such as feature transformations or attention mechanisms could be implemented. However, the primary goal of this model suite is to ensure reproducible results with 'off-the-shelf' models, which serves as a foundation for more advanced, nuanced approaches.

Figure 2: Bar plot showing best model performance for each type of machine learning model based on R2 score (though other metrics are in agreement, see Pearson r correlation and RMSE).

Figure 3: Time-series plots (left) of observed (blue) and predicted (orange) FCO2 and FCH4 fluxes for US-Myb, US-Tw1, and US-Tw4. The scatter plot (right) compares observed vs. predicted values across all sites, with a 1:1 reference line with overall and site-only performance metrics (R²).

**Figure 4:** Annual Observed and Predicted FCO2 and FCH4 Across Three Wetland Sites. Aggregated statistics for all sites are as follows: For **FCO2**, the Mean Absolute Error (MAE) is **176 gC m<sup>-2</sup> yr<sup>-1</sup>** and the Mean Bias Error (MBE) is **17 gC m<sup>-2</sup> yr<sup>-1</sup>**. For **FCH4**, MAE is **9 gC-CH<sub>4</sub> m<sup>-2</sup> yr<sup>-1</sup>** and the MBE is **1gC-CH<sub>4</sub> m<sup>-2</sup> yr<sup>-1</sup>**.

# 3.2. Model Application: Upscaling

Figure 5 displays spatial maps of annual flux estimates of Net Ecosystem Carbon Balance (NECB), and the  $CO_2$  equivalent flux rate (FCO2e) in the study domain, including zoom-in subplots highlighting areas with more data. 10 models were trained and mean and standard deviation was calculated for each spatial point. The results show that carbon sequestration, indicated by negative NECB (green) values, are typically dominant throughout the domain, although the northern regions shows more carbon sources.. In contrast, In contrast, the FCO2e distribution shows variability across the region, with sources and sinks found  $CO_2$ e sink throughout.

Figure S1 plots the coefficient of variation (CV =  $\sigma/\mu$ ) of the inter-model ensemble for both NECB and FCO2e. Higher CV indicates locations where environmental conditions are poorly represented in the training data - effectively a proxy to determine model confidence. Across the study domain the vast majority of pixels show low dispersion:  $\approx 85$  % of the mapped area has a CV < 0.5 for NECB, and 69 % falls below that same threshold for FCO2e.

Figure 6 shows averaged fluxes in the upscaling domain over the full study period. The results highlight the Delta as an overall carbon sink, with NECB averaging approximately -450 gC m<sup>-2</sup> yr<sup>-1</sup>, indicating persistent sequestration across multiple years. CH<sub>4</sub> fluxes average 31 gC-CH<sub>4</sub> m<sup>-2</sup> yr<sup>-1</sup>, and shows little spatial variability. Values are consistent with those previously reported in the region (Arias-Ortiz et al., 2021). Integrating these fluxes into a CO<sub>2</sub>-equivalent metric, this regional wetland system remains a net sink of CO<sub>2</sub> e, with approximately 600 gCO<sub>2</sub>e m<sup>-2</sup> yr<sup>-1</sup> sequestered on average in the upscaling domain, with an increasing trend with time.

Figure 5: Mean annual Net Ecosystem Carbon Balance (NECB, left) and CO<sub>2</sub>-equivalent radiative forcing (FCO<sub>2</sub>e, right) averaged over all model years. Main maps show the Delta area; dashed rectangles (1 - 3) correspond to zoom-in panels. Tidal wetlands are shaded dark blue, non-tidal light blue. Positive values (red) indicate net carbon loss; negative values (green) net uptake. See Figure 1 for reference to training sites.

Figure 6: Bar plots and box plots of annual NECB, FCH4, and FCO2e fluxes, which have been spatially integrated over the study region, a total of 25 km2 total land area vegetated primarily by Tules, but also Cattails and Phragmites. The left column shows annual fluxes for each year, with negative fluxes in green and positive fluxes in orange. Daily fluxes, aggregated to annual totals, are overlaid as grey lines. The right column shows box plots summarizing the distribution of annual fluxes, highlighting the range, median (blue line), and

spread of values. Each row represents a different flux variable: (a) NECB, (b) FCH4, and (c) FCO2e.

#### 4. Discussion

This study demonstrates the development and evaluation of a data-driven framework to upscale terrestrial CO<sub>2</sub> and CH<sub>4</sub> flux estimates for non-tidal wetlands in the Sacramento-San Joaquin Delta. By systematically comparing models of varying complexity, including linear regression, ensemble methods, gradient boosting algorithms, and recurrent neural networks (RNNs), we presented a transparent assessment of model performance. The goals were to identify the model that best predicts CO<sub>2</sub> and CH<sub>4</sub> fluxes and critically appraise whether incremental complexity is justified by improvements in predictive capacity. Relevant cited works have included many different machine learning approaches for predicting emissions. This work aims to unify modelling efforts by establishing a standard framework for developing robust data-driven models, particularly for upscaling purposes.

Our results indicate that non-linear and more advanced models generally outperformed simple linear regression approaches. Among all tested models, the Long Short-Term Memory (LSTM) and Gated Recurrent Unit (GRU) neural networks provided the highest overall skill in predicting both CO<sub>2</sub> and CH<sub>4</sub> fluxes at daily timescales. This improvement was marginal but consistent, supporting the notion that time-series models, which inherently capture temporal dependencies and non-linearities, can provide tangible benefits over linear methods and traditional machine learning algorithms.

However, while these deep learning models performed best, the performance gains were not as large as might be expected given their significantly higher complexity and computational demands. Similar outcomes have been noted in other ecological modeling applications, where advanced machine learning methods yield improvements that are statistically significant yet modest in terms of performance gains relative to linear models (Oh et al., 2022; Wood, 2022).

The deep learning models provided reasonable estimates of daily fluxes but struggled to replicate the full range of interannual variability observed in the field measurements, which is a common issue for data-driven models in this field (Nelson et al., 2024). This limited ability to capture long-term trends and extremes mirrors common challenges in machine learning-based modeling, where the absence of explicit mechanistic understanding limits extrapolation beyond the conditions represented in the training data. The difficulty in reproducing interannual

fluctuations was particularly evident for CH<sub>4</sub> fluxes, an outcome consistent with the high spatial and temporal complexity of CH<sub>4</sub> cycling in wetland environments and the limited availability of subsurface parameters (e.g., oxygen concentration, redox conditions, substrate availability) that drive CH<sub>4</sub> production. This may not be surprising as the number of annual cycles available in the training set was only 35 years.

The observed regression to the mean and the reduced dynamic range in model predictions may reflect insufficient representation of key environmental drivers in the feature set or inadequate temporal coverage and variability in the training data. While publicly available datasets such as WLDAS and LANDSATwere effective at providing spatially and temporally comprehensive inputs, the lack of direct subsurface and soil biogeochemical measurements likely limited the model's ability to capture critical internal processes that are likely causing the observed differences between years. Although the feed-forward selection process for the model features had access to an extensive pool of relevant features, results indicated that only a small subset of features was necessary to maximise performance. This suggests that, while there are many features that control CO2 and methane, their contribution to predictive accuracy may be redundant or captured indirectly by other variables. The exclusion of particular features, such as the water table depth for FCH4, illustrates the trade-off between mechanistic intuition and data-driven optimization. Strong correlations between features and limited independent variability can lead to features being left out that would typically be considered ecologically relevant.

After applying the chosen model (LSTM) to calculate CO2 and CH4 fluxes, we estimated NECB and CO<sub>2</sub>-equivalent fluxes for similar wetland settings across the Delta region. The results show spatial heterogeneity and pinpoint regions that act as stronger net carbon sinks, as well as areas where CH<sub>4</sub> emissions may offset climate benefits of net carbon sequestration. Such insights support targeted conservation and restoration strategies aimed at maximizing net carbon sequestration benefits, facilitating ongoing efforts to restore and manage wetlands to contribute to net-zero emission goals.

A key advantage of the chosen approach is its reliance on readily available, open-source data streams and standard computational resources. The framework can be deployed efficiently without specialized hardware, making it accessible to resource-limited organizations, practitioners, and researchers.

The primary objectives of this study were to identify a suitable model, contextualize model performance by comparing to a baseline linear regression, and highlight trade-offs between complexity, interpretability, and accuracy. By explicitly testing multiple models ranging from simple linear regressions to advanced recurrent neural networks, we demonstrated that complexity alone does not guarantee a substantial increase in predictive power. Instead, complexity should be adopted judiciously, based on the magnitude of performance gains, the cost of model implementation, and the level of interpretability.

We suggest that future modeling efforts should focus on deriving mechanistically relevant predictors (Ouyang et al., 2023), and incorporating hybrid modeling approaches (Yao et al., 2023) that combine the strengths of process-based and machine learning methods. Attention

mechanisms (Yuan et al., 2022), advanced architectures (e.g., Transformers (Vaswani, 2017)), or physics-informed machine learning (Raissi et al., 2019) may also help address model performance limitations.

#### 5. Conclusions

This study provides a transparent, methodical demonstration of an artificial intelligence approach to modeling wetland carbon dioxide (CO<sub>2</sub>) and methane (CH<sub>4</sub>) emissions, using a suite of "off-the-shelf" tools and establishing a standardized benchmarking protocol for model performance evaluation. In the study region (the Sacramento–San Joaquin Delta), inter-model comparisons revealed modest but appreciable performance differences when comparing advanced models with a linear regression baseline. While there are tangible benefits to employing machine learning for these purposes, it is likely that the gap between simpler models and more sophisticated models will widen as data quantity and quality continues to increase. Ultimately, this study lays the groundwork for regional scale model benchmark testing, facilitating the development of more advanced modeling approaches that can guide wetland management, restoration planning, and climate mitigation strategies.

# Code and data availability

The current version of the RCCAT model is available on GitHub at https://github.com/ashbre2/RCCAT under the MIT License. The exact version of the model used to produce the results presented in this paper has been archived on Zenodo (Brereton, 2025)

### Author contributions

Ashley developed the model, performed the analysis, and led the writing of the manuscript. Zelalem, Bhavna, Adina, William and Kunxiaojia contributed to model development and provided expertise in carbon flux modeling. Qian contributed expertise in machine learning methods. Tyler provided domain-specific knowledge of the region of interest. Yu and Yi contributed expertise in hydrological processes. All co-authors reviewed, provided input on the manuscript drafts, and approved the final version.

Abril, G. and Iversen, N.: Methane dynamics in a shallow non-tidal estuary (Randers Fjord, Denmark), Mar. Ecol. Prog. Ser., 230, 171–181, 2002.

Anderson, F. E., Bergamaschi, B., Sturtevant, C., Knox, S., Hastings, L., Windham-Myers, L., Detto, M., Hestir, E. L., Drexler, J., Miller, R. L., Matthes, J. H., Verfaillie, J., Baldocchi, D., Snyder, R. L., and Fujii, R.: Variation of energy and carbon fluxes from a restored temperate freshwater wetland and implications for carbon market verification protocols, J. Geophys. Res. Biogeosciences, 121, 777–795, https://doi.org/10.1002/2015JG003083, 2016.

Anderson, M., Gao, F., Knipper, K., Hain, C., Dulaney, W., Baldocchi, D., Eichelmann, E., Hemes, K., Yang, Y., and Medellin-Azuara, J.: Field-scale assessment of land and water use change over the California Delta using remote sensing, Remote Sens., 10, 889, 2018.

Arias-Ortiz, A., Oikawa, P. Y., Carlin, J., Masqué, P., Shahan, J., Kanneg, S., Paytan, A., and Baldocchi, D. D.: Tidal and Nontidal Marsh Restoration: A Trade-Off Between Carbon Sequestration, Methane Emissions, and Soil Accretion, J. Geophys. Res. Biogeosciences, 126, e2021JG006573, https://doi.org/10.1029/2021JG006573, 2021.

Arora, B., Wainwright, H. M., Dwivedi, D., Vaughn, L. J., Curtis, J. B., Torn, M. S., Dafflon, B., and Hubbard, S. S.: Evaluating temporal controls on greenhouse gas (GHG) fluxes in an Arctic tundra environment: An entropy-based approach, Sci. Total Environ., 649, 284–299, 2019.

Arora, B., Briggs, M. A., Zarnetske, J. P., Stegen, J., Gomez-Velez, J. D., Dwivedi, D., and Steefel, C.: Hot Spots and Hot Moments in the Critical Zone: Identification of and Incorporation into Reactive Transport Models, in: Biogeochemistry of the Critical Zone, edited by: Wymore, A. S., Yang, W. H., Silver, W. L., McDowell, W. H., and Chorover, J., Springer International Publishing, Cham, 9–47, https://doi.org/10.1007/978-3-030-95921-0\_2, 2022.

Ashley: Regional Carbon Climate Analytics Tool (RCCAT), 2025.

Aubinet, M., Vesala, T., and Papale, D.: Eddy covariance: a practical guide to measurement and data analysis, Springer Science & Business Media, 2012.

Baldocchi, D., Falge, E., Gu, L., Olson, R., Hollinger, D., Running, S., Anthoni, P., Bernhofer, C., Davis, K., and Evans, R.: FLUXNET: A new tool to study the temporal and spatial variability of ecosystem-scale carbon dioxide, water vapor, and energy flux densities, Bull. Am. Meteorol. Soc., 82, 2415–2434, 2001.

Bergmeir, C. and Benítez, J. M.: On the use of cross-validation for time series predictor evaluation, Inf. Sci., 191, 192–213, 2012.

Bernal, B. and Mitsch, W. J.: Comparing carbon sequestration in temperate freshwater wetland communities, Glob. Change Biol., 18, 1636–1647, https://doi.org/10.1111/j.1365-2486.2011.02619.x, 2012.

Bodesheim, P., Jung, M., Gans, F., Mahecha, M. D., and Reichstein, M.: Upscaled diurnal cycles of land-atmosphere fluxes: a new global half-hourly data product, Earth Syst. Sci. Data,

10, 1327-1365, 2018.

Breiman, L.: Random Forests, Mach. Learn., 45, 5–32, https://doi.org/10.1023/A:1010933404324, 2001.

Brereton, A. Regional Carbon Climate Analytics Tool (RCCAT), version 1.0.0, Zenodo, https://doi.org/10.5281/zenodo.14933820, 2025.

Brix, H., Sorrell, B. K., and Lorenzen, B.: Are Phragmites-dominated wetlands a net source or net sink of greenhouse gases?, Aquat. Bot., 69, 313–324, 2001.

Bubier, J., Costello, A., Moore, T. R., Roulet, N. T., and Savage, K.: Microtopography and methane flux in boreal peatlands, northern Ontario, Canada, Can. J. Bot., 71, 1056–1063, https://doi.org/10.1139/b93-122, 1993.

Carnell, P. E., Windecker, S. M., Brenker, M., Baldock, J., Masque, P., Brunt, K., and Macreadie, P. I.: Carbon stocks, sequestration, and emissions of wetlands in south eastern Australia, Glob. Change Biol., 24, 4173–4184, https://doi.org/10.1111/gcb.14319, 2018.

Chen, T. and Guestrin, C.: XGBoost: A Scalable Tree Boosting System, in: Proceedings of the 22nd ACM SIGKDD International Conference on Knowledge Discovery and Data Mining, KDD '16: The 22nd ACM SIGKDD International Conference on Knowledge Discovery and Data Mining, San Francisco California USA, 785–794, https://doi.org/10.1145/2939672.2939785, 2016.

Chu, H., Luo, X., Ouyang, Z., Chan, W. S., Dengel, S., Biraud, S. C., Torn, M. S., Metzger, S., Kumar, J., and Arain, M. A.: Representativeness of Eddy-Covariance flux footprints for areas surrounding AmeriFlux sites, Agric. For. Meteorol., 301, 108350, 2021.

Chung, J., Gulcehre, C., Cho, K., and Bengio, Y.: Empirical Evaluation of Gated Recurrent Neural Networks on Sequence Modeling, https://doi.org/10.48550/arXiv.1412.3555, 11 December 2014.

Ciais, P., Borges, A. V., Abril, G., Meybeck, M., Folberth, G., Hauglustaine, D., and Janssens, I. A.: The impact of lateral carbon fluxes on the European carbon balance, Biogeosciences, 5, 1259–1271, 2008.

Cortes, C.: Support-Vector Networks, Mach. Learn., 1995.

Costanza, R., De Groot, R., Sutton, P., Van der Ploeg, S., Anderson, S. J., Kubiszewski, I., Farber, S., and Turner, R. K.: Changes in the global value of ecosystem services, Glob. Environ. Change, 26, 152–158, 2014.

DeLaune, R. D. and Pezeshki, S. R.: The role of soil organic carbon in maintaining surface elevation in rapidly subsiding US Gulf of Mexico coastal marshes, Water Air Soil Pollut. Focus, 3, 167–179, 2003.

Ding, H.: Establishing a soil carbon flux monitoring system based on support vector machine and XGBoost, Soft Comput., 28, 4551–4574, 2024.

Eichelmann, E., Shortt, R., Knox, S., Sanchez, C. R., Valach, A., Sturtevant, C., Szutu, D.,

- Verfaillie, J., and Baldocchi, D.: AmeriFlux AmeriFlux US-Tw4 Twitchell East End Wetland, Lawrence Berkeley National Laboratory (LBNL), Berkeley, CA (United States ..., 2016.
- Erlingis, J. M., Rodell, M., Peters-Lidard, C. D., Li, B., Kumar, S. V., Famiglietti, J. S., Granger, S. L., Hurley, J. V., Liu, P., and Mocko, D. M.: A High-Resolution Land Data Assimilation System Optimized for the Western United States, JAWRA J. Am. Water Resour. Assoc., 57, 692–710, https://doi.org/10.1111/1752-1688.12910, 2021.
- Grande, E., Seybold, E. C., Tatariw, C., Visser, A., Braswell, A., Arora, B., Birgand, F., Haskins, J., and Zimmer, M.: Seasonal and tidal variations in hydrologic inputs drive salt marsh porewater nitrate dynamics, Hydrol. Process., 37, e14951, https://doi.org/10.1002/hyp.14951, 2023.
- Grant, R. F. and Roulet, N. T.: Methane efflux from boreal wetlands: Theory and testing of the ecosystem model Ecosys with chamber and tower flux measurements, Glob. Biogeochem. Cycles, 16, https://doi.org/10.1029/2001GB001702, 2002.
- Grant, R. F., Mekonnen, Z. A., Riley, W. J., Arora, B., and Torn, M. S.: 2. Microtopography determines how CO2 and CH4 exchange responds to changes in temperature and precipitation at an Arctic polygonal tundra site: mathematical modelling with ecosys, J Geophys Res Biogeosci, 122, 3174–3187, 2017.
- Harrison, R. B., Footen, P. W., and Strahm, B. D.: Deep soil horizons: contribution and importance to soil carbon pools and in assessing whole-ecosystem response to management and global change, For. Sci., 57, 67–76, 2011.
- Hastie, T.: The elements of statistical learning: data mining, inference, and prediction, 2009.
- Hemes, K. S., Chamberlain, S. D., Eichelmann, E., Anthony, T., Valach, A., Kasak, K., Szutu, D., Verfaillie, J., Silver, W. L., and Baldocchi, D. D.: Assessing the carbon and climate benefit of restoring degraded agricultural peat soils to managed wetlands, Agric. For. Meteorol., 268, 202–214, 2019.
- Hill, T., Chocholek, M., and Clement, R.: The case for increasing the statistical power of eddy covariance ecosystem studies: why, where and how?, Glob. Change Biol., 23, 2154–2165, https://doi.org/10.1111/gcb.13547, 2017.
- Hochreiter, S.: Long Short-term Memory, Neural Comput. MIT-Press, 1997.
- Huete, A. R.: A soil-adjusted vegetation index (SAVI), Remote Sens. Environ., 25, 295–309, 1988.
- Justice, C. O., Townshend, J. R. G., Vermote, E. F., Masuoka, E., Wolfe, R. E., Saleous, N., Roy, D. P., and Morisette, J. T.: An overview of MODIS Land data processing and product status, Remote Sens. Environ., 83, 3–15, 2002.
- Kaufman, S., Rosset, S., Perlich, C., and Stitelman, O.: Leakage in data mining: Formulation, detection, and avoidance, ACM Trans. Knowl. Discov. Data, 6, 1–21, https://doi.org/10.1145/2382577.2382579, 2012.
- Ke, G., Meng, Q., Finley, T., Wang, T., Chen, W., Ma, W., Ye, Q., and Liu, T.-Y.: Lightgbm: A

- highly efficient gradient boosting decision tree, Adv. Neural Inf. Process. Syst., 30, 2017.
- de Klein, J. J. and van der Werf, A. K.: Balancing carbon sequestration and GHG emissions in a constructed wetland, Ecol. Eng., 66, 36–42, 2014.
- Knox, S. H., Sturtevant, C., Matthes, J. H., Koteen, L., Verfaillie, J., and Baldocchi, D.: Agricultural peatland restoration: effects of land-use change on greenhouse gas (CO2 and CH4) fluxes in the Sacramento-San Joaquin Delta, Glob. Change Biol., 21, 750–765, https://doi.org/10.1111/gcb.12745, 2015.
- Knox, S. H., Dronova, I., Sturtevant, C., Oikawa, P. Y., Matthes, J. H., Verfaillie, J., and Baldocchi, D.: Using digital camera and Landsat imagery with eddy covariance data to model gross primary production in restored wetlands, Agric. For. Meteorol., 237, 233–245, 2017.
- Knox, S. H., Bansal, S., McNicol, G., Schafer, K., Sturtevant, C., Ueyama, M., Valach, A. C., Baldocchi, D., Delwiche, K., Desai, A. R., Euskirchen, E., Liu, J., Lohila, A., Malhotra, A., Melling, L., Riley, W., Runkle, B. R. K., Turner, J., Vargas, R., Zhu, Q., Alto, T., Fluet-Chouinard, E., Goeckede, M., Melton, J. R., Sonnentag, O., Vesala, T., Ward, E., Zhang, Z., Feron, S., Ouyang, Z., Alekseychik, P., Aurela, M., Bohrer, G., Campbell, D. I., Chen, J., Chu, H., Dalmagro, H. J., Goodrich, J. P., Gottschalk, P., Hirano, T., Iwata, H., Jurasinski, G., Kang, M., Koebsch, F., Mammarella, I., Nilsson, M. B., Ono, K., Peichl, M., Peltola, O., Ryu, Y., Sachs, T., Sakabe, A., Sparks, J. P., Tuittila, E., Vourlitis, G. L., Wong, G. X., Windham-Myers, L., Poulter, B., and Jackson, R. B.: Identifying dominant environmental predictors of freshwater wetland methane fluxes across diurnal to seasonal time scales, Glob. Change Biol., 27, 3582–3604, https://doi.org/10.1111/gcb.15661, 2021.
- Kumar, A., Bhatia, A., Fagodiya, R. K., Malyan, S. K., and Meena, B. L.: Eddy covariance flux tower: A promising technique for greenhouse gases measurement, Adv Plants Agric Res, 7, 337–340, 2017.
- Laćan, I. and Resh, V. H.: A case study in integrated management: Sacramento–San Joaquin Rivers and Delta of California, USA, Ecohydrol. Hydrobiol., 16, 215–228, 2016.
- Landsat, U.: Landsat 8-9Operational Land Imager (OLI)-Thermal Infrared Sensor (TIRS) Collection 2 Level 2 (L2) Data Format Control Book (DFCB), U. S. Geol. Surv. Rest. VA USA, 78, 2020.
- Lee, H., Calvin, K., Dasgupta, D., Krinner, G., Mukherji, A., Thorne, P., Trisos, C., Romero, J., Aldunce, P., and Barrett, K.: Climate change 2023: synthesis report. Contribution of working groups I, II and III to the sixth assessment report of the intergovernmental panel on climate change, The Australian National University, 2023.
- Li, C.: The DNDC Model, in: Evaluation of Soil Organic Matter Models, edited by: Powlson, D. S., Smith, P., and Smith, J. U., Springer Berlin Heidelberg, Berlin, Heidelberg, 263–267, https://doi.org/10.1007/978-3-642-61094-3\_20, 1996.
- Lolu, A. J., Ahluwalia, A. S., Sidhu, M. C., Reshi, Z. A., and Mandotra, S. K.: Carbon Sequestration and Storage by Wetlands: Implications in the Climate Change Scenario, in: Restoration of Wetland Ecosystem: A Trajectory Towards a Sustainable Environment, edited by: Upadhyay, A. K., Singh, R., and Singh, D. P., Springer Singapore, Singapore, 45–58, https://doi.org/10.1007/978-981-13-7665-8\_4, 2020.

- López, D., Sepúlveda, M., and Vidal, G.: Phragmites australis and Schoenoplectus californicus in constructed wetlands: Development and nutrient uptake, J. Soil Sci. Plant Nutr., 16, 763–777, 2016.
- Lund, J., Hanak, E., Fleenor, W., Bennett, W., and Howitt, R.: Comparing futures for the Sacramento, San Joaquin delta, Univ of California Press, 2010.
- Mack, S. K., Lane, R. R., Deng, J., Morris, J. T., and Bauer, J. J.: Wetland carbon models: Applications for wetland carbon commercialization, Ecol. Model., 476, 110228, 2023.
- Matthes, J. H., Sturtevant, C., Oikawa, P., Chamberlain, S. D., Szutu, D., Arias-Ortiz, A., Verfaillie, J., and Baldocchi, D.: AmeriFlux AmeriFlux US-Myb Mayberry Wetland, Lawrence Berkeley National Laboratory (LBNL), Berkeley, CA (United States ..., 2016.
- Meyer, H. and Pebesma, E.: Machine learning-based global maps of ecological variables and the challenge of assessing them, Nat. Commun., 13, 2208, 2022.
- Miller, R. L., Fram, M., Fujii, R., and Wheeler, G.: Subsidence reversal in a re-established wetland in the Sacramento-San Joaquin Delta, California, USA, San Franc. Estuary Watershed Sci., 6, 2008.
- Mitsch, W. J. and Gosselink, J. G.: Wetlands, John wiley & sons, 2015a.
- Mitsch, W. J. and Gosselink, J. G.: Wetlands, John wiley & sons, 2015b.
- Mitsch, W. J., Bernal, B., Nahlik, A. M., Mander, Ü., Zhang, L., Anderson, C. J., Jørgensen, S. E., and Brix, H.: Wetlands, carbon, and climate change, Landsc. Ecol., 28, 583–597, 2013.
- Moomaw, W. R., Chmura, G. L., Davies, G. T., Finlayson, C. M., Middleton, B. A., Natali, S. M., Perry, J. E., Roulet, N., and Sutton-Grier, A. E.: Wetlands In a Changing Climate: Science, Policy and Management, Wetlands, 38, 183–205, https://doi.org/10.1007/s13157-018-1023-8, 2018.
- Nahlik, A. M. and Fennessy, M. S.: Carbon storage in US wetlands, Nat. Commun., 7, 1–9, 2016.
- Nelson, J. A., Walther, S., Gans, F., Kraft, B., Weber, U., Novick, K., Buchmann, N., Migliavacca, M., Wohlfahrt, G., and Šigut, L.: X-BASE: the first terrestrial carbon and water flux products from an extended data-driven scaling framework, FLUXCOM-X, Biogeosciences, 21, 5079–5115, 2024.
- Novick, K. A., Biederman, J. A., Desai, A. R., Litvak, M. E., Moore, D. J., Scott, R. L., and Torn, M. S.: The AmeriFlux network: A coalition of the willing, Agric. For. Meteorol., 249, 444–456, 2018.
- Oh, M., Lee, J., Kim, J., and Kim, H.: Machine learning-based statistical downscaling of wind resource maps using multi-resolution topographical data, Wind Energy, 25, 1121–1141, https://doi.org/10.1002/we.2718, 2022.
- Ouyang, Z., Jackson, R. B., McNicol, G., Fluet-Chouinard, E., Runkle, B. R., Papale, D., Knox, S. H., Cooley, S., Delwiche, K. B., and Feron, S.: Paddy rice methane emissions across

- Monsoon Asia, Remote Sens. Environ., 284, 113335, 2023.
- Parton, W. J., Hartman, M., Ojima, D., and Schimel, D.: DAYCENT and its land surface submodel: description and testing, Glob. Planet. Change, 19, 35–48, 1998.
- Pastorello, G., Trotta, C., Canfora, E., Chu, H., Christianson, D., Cheah, Y.-W., Poindexter, C., Chen, J., Elbashandy, A., and Humphrey, M.: The FLUXNET2015 dataset and the ONEFlux processing pipeline for eddy covariance data, Sci. Data, 7, 225, 2020.
- Peltola, O., Vesala, T., Gao, Y., Räty, O., Alekseychik, P., Aurela, M., Chojnicki, B., Desai, A. R., Dolman, A. J., and Euskirchen, E. S.: Monthly gridded data product of northern wetland methane emissions based on upscaling eddy covariance observations, Earth Syst. Sci. Data, 11, 1263–1289, 2019.
- Perry, G. L. W., Seidl, R., Bellvé, A. M., and Rammer, W.: An Outlook for Deep Learning in Ecosystem Science, Ecosystems, 25, 1700–1718, https://doi.org/10.1007/s10021-022-00789-y, 2022.
- Raissi, M., Perdikaris, P., and Karniadakis, G. E.: Physics-informed neural networks: A deep learning framework for solving forward and inverse problems involving nonlinear partial differential equations, J. Comput. Phys., 378, 686–707, 2019.
- Räsänen, A., Manninen, T., Korkiakoski, M., Lohila, A., and Virtanen, T.: Predicting catchment-scale methane fluxes with multi-source remote sensing, Landsc. Ecol., 36, 1177–1195, https://doi.org/10.1007/s10980-021-01194-x, 2021.
- Rey-Sanchez, C., Wharton, S., Vilà-Guerau De Arellano, J., Paw U, K. T., Hemes, K. S., Fuentes, J. D., Osuna, J., Szutu, D., Ribeiro, J. V., Verfaillie, J., and Baldocchi, D.: Evaluation of Atmospheric Boundary Layer Height From Wind Profiling Radar and Slab Models and Its Responses to Seasonality of Land Cover, Subsidence, and Advection, J. Geophys. Res. Atmospheres, 126, e2020JD033775, https://doi.org/10.1029/2020JD033775, 2021.
- Saunois, M., Martinez, A., Poulter, B., Zhang, Z., Raymond, P., Regnier, P., Canadell, J. G., Jackson, R. B., Patra, P. K., and Bousquet, P.: Global Methane Budget 2000–2020, Earth Syst. Sci. Data Discuss., 2024, 1–147, 2024.
- Sharma, S. and Singh, P.: Wetlands conservation: Current challenges and future strategies, John Wiley & Sons, 2021.
- Stewart, G. A., Sharp, S. J., Taylor, A. K., Williams, M. R., and Palmer, M. A.: High spatial variability in wetland methane fluxes is tied to vegetation patch types, Biogeochemistry, https://doi.org/10.1007/s10533-024-01188-2, 2024.
- Tashman, L. J.: Out-of-sample tests of forecasting accuracy: an analysis and review, Int. J. Forecast., 16, 437–450, 2000.
- Tramontana, G., Jung, M., Schwalm, C. R., Ichii, K., Camps-Valls, G., Ráduly, B., Reichstein, M., Arain, M. A., Cescatti, A., and Kiely, G.: Predicting carbon dioxide and energy fluxes across global FLUXNET sites with regression algorithms, Biogeosciences, 13, 4291–4313, 2016.
- Uhran, B. R., Windham-Myers, L., Bliss, N. B., Nahlik, A., Sundquist, E. T., and Stagg, C. L.:

Harmonizing wetland soil organic carbon datasets to improve spatial representation of 2011 soil carbon stocks in the conterminous United States, https://doi.org/10.5066/P9H1PIX3, 2022.

Upadhyay, A. K., Singh, R., and Singh, D. P. (Eds.): Restoration of Wetland Ecosystem: A Trajectory Towards a Sustainable Environment, Springer Singapore, Singapore, https://doi.org/10.1007/978-981-13-7665-8, 2020.

Valach, A., Shortt, R., Szutu, D., Eichelmann, E., Knox, S., Hemes, K., Verfaillie, J., and Baldocchi, D.: AmeriFlux AmeriFlux US-Tw1 Twitchell Wetland West Pond, Lawrence Berkeley National Laboratory (LBNL), Berkeley, CA (United States ..., 2016.

Vaswani, A.: Attention is all you need, Adv. Neural Inf. Process. Syst., 2017.

Villa, J. A. and Bernal, B.: Carbon sequestration in wetlands, from science to practice: An overview of the biogeochemical process, measurement methods, and policy framework, Ecol. Eng., 114, 115–128, 2018.

Weiler, D. A., Tornquist, C. G., Zschornack, T., Ogle, S. M., Carlos, F. S., and Bayer, C.: Daycent simulation of methane emissions, grain yield, and soil organic carbon in a subtropical paddy rice system, Rev. Bras. Ciênc. Solo, 42, e0170251, 2018.

Whiting, G. J. and Chanton, J. P.: Primary production control of methane emission from wetlands, Nature, 364, 794–795, 1993.

Windham-Myers, L., Bergamaschi, B., Anderson, F., Knox, S., Miller, R., and Fujii, R.: Potential for negative emissions of greenhouse gases (CO2, CH4 and N2O) through coastal peatland reestablishment: Novel insights from high frequency flux data at meter and kilometer scales, Environ. Res. Lett., 13, 045005, 2018.

Wood, D. A.: Machine learning and regression analysis reveal different patterns of influence on net ecosystem exchange at two conifer woodland sites, Res. Ecol., 4, 24–50, 2022.

Workgroup, C. W. M.: EcoAtlas, 2019.

Xu, X. and Trugman, A. T.: Trait-Based Modeling of Terrestrial Ecosystems: Advances and Challenges Under Global Change, Curr. Clim. Change Rep., 7, 1–13, https://doi.org/10.1007/s40641-020-00168-6, 2021.

Xu, Y., Liu, X., Cao, X., Huang, C., Liu, E., Qian, S., Liu, X., Wu, Y., Dong, F., and Qiu, C.-W.: Artificial intelligence: A powerful paradigm for scientific research, The Innovation, 2, 2021.

Yao, S., Chen, C., Chen, Q., Zhang, J., and He, M.: Combining process-based model and machine learning to predict hydrological regimes in floodplain wetlands under climate change, J. Hydrol., 626, 130193, 2023.

Yin, X., Jiang, C., Xu, S., Yu, X., Yin, X., Wang, J., Maihaiti, M., Wang, C., Zheng, X., and Zhuang, X.: Greenhouse gases emissions of constructed wetlands: mechanisms and affecting factors, Water, 15, 2871, 2023.

Yuan, K., Zhu, Q., Li, F., Riley, W. J., Torn, M., Chu, H., McNicol, G., Chen, M., Knox, S., and Delwiche, K.: Causality guided machine learning model on wetland CH4 emissions across

global wetlands, Agric. For. Meteorol., 324, 109115, 2022.

Yuan, K., Li, F., McNicol, G., Chen, M., Hoyt, A., Knox, S., Riley, W. J., Jackson, R., and Zhu, Q.: Boreal–Arctic wetland methane emissions modulated by warming and vegetation activity, Nat. Clim. Change, 14, 282–288, 2024.

Yvon-Durocher, G., Allen, A. P., Bastviken, D., Conrad, R., Gudasz, C., St-Pierre, A., Thanh-Duc, N., and Del Giorgio, P. A.: Methane fluxes show consistent temperature dependence across microbial to ecosystem scales, Nature, 507, 488–491, 2014.

Zhang, Y., Li, C., Trettin, C. C., Li, H., and Sun, G.: An integrated model of soil, hydrology, and vegetation for carbon dynamics in wetland ecosystems, Glob. Biogeochem. Cycles, 16, https://doi.org/10.1029/2001GB001838, 2002.

Zou, H., Chen, J., Li, X., Abraha, M., Zhao, X., and Tang, J.: Modeling net ecosystem exchange of CO2 with gated recurrent unit neural networks, Agric. For. Meteorol., 350, 109985, 2024.

### Supplementary

#### Carbon Sequestration in Nontidal Wetlands

Various studies have documented substantial rates of carbon uptake in nontidal wetlands. Table 1 summarizes reported carbon sequestration rates from prominent publications.

Table S1: Reported Carbon Sequestration Rates in Nontidal Wetlands

| Location                                             | Method                                                 | Climate           | Scale<br>(# of<br>Sites) | Descriptors                                                                  | Sequestration<br>Rate<br>(g C/m²/year) | Cited<br>Study                                                |
|------------------------------------------------------|--------------------------------------------------------|-------------------|--------------------------|------------------------------------------------------------------------------|----------------------------------------|---------------------------------------------------------------|
| San<br>Francisco<br>Bay-Delta<br>(Young<br>Wetlands) | Soil coring, 210Pb radiometric dating, eddy covariance | Mediterran<br>ean | 1 site                   | Nontidal<br>managed<br>wetland<br>dominated by<br>Typha spp.,<br>Phragmites  | 334 ± 70                               | Arias-Ortiz<br>et al. (2021)<br>(Arias-Ortiz<br>et al., 2021) |
| San<br>Francisco<br>Bay-Delta<br>(Old<br>Wetlands)   | Soil coring, 210Pb radiometric dating, eddy covariance | Mediterran<br>ean | 1 site                   | Nontidal<br>managed<br>wetland<br>dominated by<br>dense Typha<br>spp. canopy | 357 ± 102                              | Arias-Ortiz<br>et al. (2021)<br>(Arias-Ortiz<br>et al., 2021) |

| Central Ohio<br>(Gahanna<br>Woods)       | 137Cs and 210Pb radiometric dating | Temperate | 1 site,<br>small<br>scale      | Depressional<br>wetland -Shrub<br>dominated by<br>Cephalanthus<br>occidentalis | 202 | Bernal and<br>Mitsch<br>(2012)<br>(Bernal and<br>Mitsch,<br>2012) |
|------------------------------------------|------------------------------------|-----------|--------------------------------|--------------------------------------------------------------------------------|-----|-------------------------------------------------------------------|
| Central Ohio<br>(Gahanna<br>Woods)       | 137Cs and 210Pb radiometric dating | Temperate | 1 site,<br>small<br>scale      | Depressional<br>wetland -<br>Forested<br>dominated by<br>Quercus<br>palustris  | 473 | Bernal and<br>Mitsch<br>(2012)(Bern<br>al and<br>Mitsch,<br>2012) |
| Central Ohio<br>(Gahanna<br>Woods)       | 137Cs and 210Pb radiometric dating | Temperate | 1 site,<br>small<br>scale      | Depressional<br>wetland -Marsh<br>dominated by<br>Typha spp.                   | 210 | Bernal and<br>Mitsch<br>(2012)(Bern<br>al and<br>Mitsch,<br>2012) |
| Northern<br>Ohio (Old<br>Woman<br>Creek) | 137Cs and 210Pb radiometric dating | Temperate | 1 site,<br>mediu<br>m<br>scale | Riverine wetland -Marsh dominated by Phragmites australis, Scirpus fluviatilis | 105 | Bernal and<br>Mitsch<br>(2012)(Bern<br>al and<br>Mitsch,<br>2012) |
| Northern<br>Ohio (Old<br>Woman<br>Creek) | 137Cs and 210Pb radiometric dating | Temperate | 1 site,<br>mediu<br>m<br>scale | Riverine wetland - Mudflat dominated by Leersia oryzoides                      | 112 | Bernal and<br>Mitsch<br>(2012)(Bern<br>al and<br>Mitsch,<br>2012) |
| Northern<br>Ohio (Old<br>Woman<br>Creek) | 137Cs and 210Pb radiometric dating | Temperate | 1 site,<br>mediu<br>m<br>scale | Riverine wetland - Floating bed dominated by Nelumbo lutea                     | 160 | Bernal and<br>Mitsch<br>(2012)(Bern<br>al and<br>Mitsch,<br>2012) |

| Victoria,<br>Australia | Core sampling,<br>model (Appleby &<br>Oldfield, 1978;<br>Krishnaswami,<br>Lal, Martin, &<br>Meybeck, 1971) | Temperate               | 19<br>sites  | Shallow<br>freshwater<br>marsh:<br>moderate<br>carbon stocks  | 91            | Carnell et<br>al.<br>(2018)(Car<br>nell et al.,<br>2018)                            |
|------------------------|------------------------------------------------------------------------------------------------------------|-------------------------|--------------|---------------------------------------------------------------|---------------|-------------------------------------------------------------------------------------|
| Victoria,<br>Australia | Core sampling,<br>model (Appleby &<br>Oldfield, 1978;<br>Krishnaswami,<br>Lal, Martin, &<br>Meybeck, 1971) | Temperate               | 22<br>sites  | Permanent<br>open freshwater<br>wetlands: low<br>carbon stock | 230           | Carnell et<br>al.<br>(2018)(Car<br>nell et al.,<br>2018)                            |
| Victoria,<br>Australia | Core sampling,<br>model (Appleby &<br>Oldfield, 1978;<br>Krishnaswami,<br>Lal, Martin, &<br>Meybeck, 1971) | Temperate               | 33<br>sites  | Deep freshwater<br>marsh: high<br>carbon stocks               | 160           | Carnell et<br>al.<br>(2018)(Car<br>nell et al.,<br>2018)                            |
| Netherlands            | Biomass<br>measurement                                                                                     | Temperate               | 1 site       | Constructed wetland with emergent vegetation (Phragmites)     | 797 (average) | De Klein<br>and van der<br>Werf<br>(2014)(de<br>Klein and<br>van der<br>Werf, 2014) |
| Global                 | Soil coring                                                                                                | Temperate<br>& tropical | 7 sites      | Includes natural<br>and created<br>wetlands                   | 118 (average) | Mitsch et al.<br>(2013)(Mits<br>ch et al.,<br>2013)                                 |
| Global                 | Marker horizons,<br>137Cs and 210Pb<br>radiometric dating                                                  | Temperate<br>/Tropical  | 186<br>sites | Inland wetland -<br>Permanent<br>Freshwater<br>Marsh          | 122.6         | Villa and<br>Bernal<br>(2018)(Villa<br>and Bernal,<br>2018)                         |
| Global                 | Radiometric<br>dating (14C)                                                                                | Temperate<br>/Boreal    | 88<br>sites  | Rain-fed<br>bogs/mires -<br>Non-forested<br>Peatland          | 26.1          | Villa and<br>Bernal<br>(2018)(Villa<br>and Bernal,<br>2018)                         |

| Global | Dendrogeomor-<br>phic techniques,<br>14C and 210Pb<br>radiometric dating | Temperate<br>/Tropical in<br>riparian<br>settings | 117<br>sites | Riparian/Bottom<br>land Forests -<br>Freshwater<br>Tree-Dominated<br>Wetland | 176 | Villa and<br>Bernal<br>(2018)(Villa<br>and Bernal,<br>2018) |
|--------|--------------------------------------------------------------------------|---------------------------------------------------|--------------|------------------------------------------------------------------------------|-----|-------------------------------------------------------------|
|--------|--------------------------------------------------------------------------|---------------------------------------------------|--------------|------------------------------------------------------------------------------|-----|-------------------------------------------------------------|

## Methane Emissions in Nontidal Wetlands

While nontidal wetlands sequester carbon, they can also emit methane, potentially offsetting some climate mitigation benefits. Table 2 presents methane emission rates from various studies.

Table S2: Reported Methane Emissions in Nontidal Wetlands

| Location                                          | Method                                                                                | Climate              | Scale<br>(# of<br>Sites) | Descriptors                                                                                           | Methane<br>Emissions<br>(g C-CH4<br>m-2 yr-1) | Cited<br>Study                                                       |
|---------------------------------------------------|---------------------------------------------------------------------------------------|----------------------|--------------------------|-------------------------------------------------------------------------------------------------------|-----------------------------------------------|----------------------------------------------------------------------|
| San Francisco<br>Bay-Delta<br>(Young<br>Wetlands) | Eddy covariance                                                                       | Mediterran<br>ean    | 1 site                   | Nontidal managed<br>wetland<br>dominated by<br>Typha spp.,<br>Phragmites                              | 44 ± 5                                        | Arias-Ortiz<br>et al.<br>(2021)<br>(Arias-<br>Ortiz et al.,<br>2021) |
| San Francisco<br>Bay-Delta (Old<br>Wetlands)      | Eddy covariance                                                                       | Mediterran<br>ean    | 1 site                   | Nontidal managed<br>wetland<br>dominated by<br>dense Typha spp.<br>canopy                             | 37 ± 4                                        | Arias-Ortiz<br>et al.<br>(2021)<br>(Arias-<br>Ortiz et al.,<br>2021) |
| Maryland                                          | Static chambers<br>and eddy<br>covariance<br>(combined in a<br>Bayesian<br>framework) | Humid<br>subtropical | 1 site                   | Restored<br>freshwater<br>wetlands with<br>graminoid patches<br>dominated by<br>grasses and<br>sedges | ~142<br>(median)                              | Stewart et<br>al. (2024)<br>(Stewart et<br>al., 2024)                |

| Delmarva<br>Peninsula,<br>Maryland | Static chambers<br>and eddy<br>covariance<br>(combined in a<br>Bayesian<br>framework)                                             | Humid<br>subtropical | 1 site     | Open water areas                 | ~5                 | Stewart et<br>al. (2024)<br>(Stewart et<br>al., 2024)                      |
|------------------------------------|-----------------------------------------------------------------------------------------------------------------------------------|----------------------|------------|----------------------------------|--------------------|----------------------------------------------------------------------------|
| Louisiana                          | Gas diffusion<br>chambers                                                                                                         | Humid<br>subtropical | 3<br>sites | Freshwater<br>marshes            | 3–225              | Delaune<br>and<br>Pezeshki<br>(2003)(De<br>Laune and<br>Pezeshki,<br>2003) |
| Ohio                               | Non-steady state<br>gas sampling<br>chamber method<br>(Altor and Mitsch<br>(2006, 2008) and<br>Nahlik and Mitsch<br>(2010, 2011)) | Temperate            | 1 site     | Natural wetland                  | 57                 | Mitsch et<br>al.<br>(2013)(Mit<br>sch et al.,<br>2013)                     |
| Ohio                               | Non-steady state<br>gas sampling<br>chamber method<br>(Altor and Mitsch<br>(2006, 2008) and<br>Nahlik and Mitsch<br>(2010, 2011)) | Temperate            | 2<br>sites | Created marshes                  | 30                 | Mitsch et<br>al.<br>(2013)(Mit<br>sch et al.,<br>2013)                     |
| Costa Rica                         | Non-steady state<br>gas sampling<br>chamber method<br>(Altor and Mitsch<br>(2006, 2008) and<br>Nahlik and Mitsch<br>(2010, 2011)) | Tropical             | 3<br>sites | Isolated & floodplain wetlands   | Highest<br>220–263 | Mitsch et<br>al.<br>(2013)(Mit<br>sch et al.,<br>2013)                     |
| Costa Rica                         | Non-steady state<br>gas sampling<br>chamber method<br>(Altor and Mitsch<br>(2006, 2008) and<br>Nahlik and Mitsch<br>(2010, 2011)) | Tropical             | 1 site     | Flow-through<br>tropical wetland | 33                 | Mitsch et<br>al.<br>(2013)(Mit<br>sch et al.,<br>2013)                     |

| Randers<br>Fjord,<br>Denmark | Sediment core   | Temperate                                         | 1 site      | Estuarine<br>Environments<br>(Freshwater<br>Zones) | 2.08     | Abril and<br>Iversen<br>(2002)(Abr<br>il and<br>Iversen,<br>2002) |
|------------------------------|-----------------|---------------------------------------------------|-------------|----------------------------------------------------|----------|-------------------------------------------------------------------|
| Randers<br>Fjord,<br>Denmark | Sediment core   | Temperate                                         | 1 site      | Estuarine<br>Environments<br>(Saltwater Zones)     | 0.23     | Abril and<br>Iversen<br>(2002)(Abr<br>il and<br>Iversen,<br>2002) |
| Global                       | Eddy covariance | Boreal,<br>temperate,<br>tropical/su<br>btropical | 23<br>sites | Freshwater<br>Wetlands                             | 0.25–271 | Knox et al.<br>(2021)(Kn<br>ox et al.,<br>2021)                   |

Table S3: Feature pool

| Variable Name                                                  | Full Variable Name            | Variable Name | Full Variable Name                    |  |  |  |
|----------------------------------------------------------------|-------------------------------|---------------|---------------------------------------|--|--|--|
| WLDAS variables: https://ldas.gsfc.nasa.gov/wldas/model-output |                               |               |                                       |  |  |  |
| AvgSurfT_tavg                                                  | Surface<br>Temperature        | Rainf_f_tavg  | Rainfall Flux (Rain + Snow)           |  |  |  |
| BareSoilT_tavg                                                 | Bare Soil<br>Temperature      | Rainf_tavg    | Precipitation Rate                    |  |  |  |
| CanopInt_tavg                                                  | Total Canopy Water<br>Storage | SWdown_f_tavg | Surface Downwelling<br>Shortwave Flux |  |  |  |

| ECanop_tavg                                                        | Interception<br>Evaporation                  | Soil Moisture        | Soil Moisture (0-200 cm), m <sup>3</sup> m <sup>-3</sup> |  |  |
|--------------------------------------------------------------------|----------------------------------------------|----------------------|----------------------------------------------------------|--|--|
| ESoil_tavg                                                         | Bare Soil<br>Evaporation                     | Soil Temperature     | Soil Temperature (0-100 cm),<br>K                        |  |  |
| Evap_tavg                                                          | Total<br>Evapotranspiration                  | Swnet_tavg           | Surface Net Downward<br>Shortwave Flux                   |  |  |
| LWdown_f_tavg                                                      | Surface<br>Downwelling<br>Longwave Flux      | TVeg_tavg            | Vegetation Transpiration                                 |  |  |
| Lwnet_tavg                                                         | Surface Net<br>Downward<br>Longwave Flux     | Tair_f_tavg          | Air Temperature                                          |  |  |
| Psurf_f_tavg                                                       | Surface Pressure                             | VegT_tavg            | Canopy Temperature                                       |  |  |
| Qair_f_tavg                                                        | Specific Humidity                            | Wind_f_tavg          | Wind Speed                                               |  |  |
| Qg_tavg                                                            | Downward Heat Flux<br>in Soil                | WT_tavg              | Water in Aquifer and Saturated<br>Soil                   |  |  |
| Qh_tavg                                                            | Surface Upward<br>Sensible Heat Flux         | WaterTableD_tav<br>g | Water Table Depth                                        |  |  |
| Qle_tavg                                                           | Surface Upward<br>Latent Heat Flux           | Qs_tavg              | Surface Runoff Amount                                    |  |  |
| LANDSAT variables: https://landsat.gsfc.nasa.gov/data/data-access/ |                                              |                      |                                                          |  |  |
| NDVI                                                               | Normalized<br>Difference<br>Vegetation Index | EVI                  | Enhanced Vegetation Index                                |  |  |

| SAVI  | Soil-Adjusted<br>Vegetation Index                | NDWI | Normalized Difference Water Index        |
|-------|--------------------------------------------------|------|------------------------------------------|
| NDMI  | Normalized Difference Moisture Index             | NDGI | Normalized Difference<br>Greenness Index |
| MNDWI | Modified Normalized<br>Difference Water<br>Index |      |                                          |

Figure S1: Coefficient of variation  $(\sigma/|\mu|)$  of NECB (left) and FCO2e (right) across the 10 runs, showing relative inter-model uncertainty in the predictions. Green areas show high model confidence and red areas shower either lower model confidence (or division by small  $\mu$ ). This can be interpreted as a proxy for the confidence of the spatial upscaling.