# Peer review of "To be submitted: Geoscientific Model Development"

_EGUsphere, 2025_

## Author Comment (AC1)

**Response to Reviewers**

**Manuscript**:
Development of a Model Framework for Terrestrial Carbon Flux Prediction: the Regional Carbon and Climate Analytics Tool (RCCAT) Applied to Non-tidal Wetlands

We thank and appreciate the reviewers for their positive and constructive feedback. and have incorporated the following improvements:

- Colour-blind palette across all figures, every plot now uses a Python colour-blind friendly scheme, making lines and markers clearly distinguishable for all readers.

- Mapped-figure overhaul – Figure 5 (and the matching base map in Figure 1) now includes three zoom-in panels on the most data-dense areas, plus tidal vs non-tidal overlays, so local variability is easier to inspect without losing regional context. We also updated Figure 3's plotting style for better visual.

- Detailed data-processing description – Section 2.3 has been expanded to give step-by-step information on data processing steps.

- MODIS → LANDSAT swap: Higher spatial resolution LANDSAT indices replace MODIS products after head-to-head tests showed noticeably better model skill with fewer predictors.

- Clearer feature interpretation – Updated text to include mechanistic process interpretation.

**Reviewer #1**

**Comment 1**

"Line 381: After selecting LSTM as the model of choice…This paragraph belongs to the Methods as it explains how the work is done with very little with the actual results."

**Response**. We agree and have moved this paragraph from the Results section to the end of Section 2.6.

**Manuscript changes**. Paragraph relocated to Methods 2.6

**Comment 2**

"Figure 3: The lines in the legends here need to be thicker as in its current presentation, it is very difficult, at least for me, to gather with a quick glance which color represents which line. Additionally I would recommend reconsidering using, for example, red and blue instead of blue and green as the shades applied here are a bit too close to each other."

**Response**. We redesigned Figure 3 as a scatter plot, increased marker and line thickness, and adopted a colour-blind safe palette in Python to avoid issues. We also updated all color schemes in the other figures.

**Manuscript changes**. Updated Figure 3 graphic. Furthermore, the 14-day smoothing that was applied to target and features was removed for clarity (see reviewer #2 Comment 3)

**Comment 3**

"Figure 5: This figure should just be moved to supplemental material. There is just far too much empty space here—some of the locations with data in it are so small that I had to look at the figure for a long while to be certain if it was even there. Note that while I am critical of this, I also cannot think of a better way to visually present this kind of map data."

**Response**. We do not want to move the figure to SI because we want to showcase upscaling of results from our study. In response to this comment, we have now redesigned Figure 5 for clarity. The new composite includes three zoom-in panels focused on data-dense regions and overlays tidal vs. non-tidal wetland polygons to match with updated Figure 1.

**Manuscript changes**. Main Figure 5 replaced with redesigned composite; updated caption describing zoom-ins and wetland classification overlay. Figure one changed to just include tidal and nontidal areas for clarity.

**Reviewer #2**

**Comment 1** - Lines 19–21: "Since this study focuses on daily-scale calculations and a specific region, the computational cost is weak as the limitations of the process-based model. It would be more appropriate to discuss the constraints of the input parameters, as mentioned in the introduction."

**Response**. Agreed. The sentence has been rewritten to emphasise that site-specific parameterisation and data availability are the primary limitations, we removed computational considerations as both models can be run on standard laptops.

**Manuscript changes**. Revised sentence in Abstract.

**Comment 2**

"I assume that the scores presented in this paragraph were derived from LOSO cross-validation results across all three sites, as shown in the scatter plots in Figure 3. However, due to the lack of information on how these scores were calculated, it is difficult for readers to determine whether they support the validity of the upscaling and extrapolation."

**Response**. Section 2.6 now specifies that $R^2$, RMSE, and Pearson's r are computed for each held-out site and then recomputed on the pooled set of all held-out predictions. Figure 3 displays both per-site metrics (in the legend) and the overall pooled metrics.

**Manuscript changes**. Updated explanatory text to Methods 2.6; updated Figure 3.

**Comment 3**

"The scatter plots in Fig.3 show a patterned, line-like distribution. Is there a specific reason for this? While it might be due to regression to the mean, could this also occur with data that includes observational errors?"

**Response**. The striping is an artefact of the 14-day running mean smoothing applied to targets and features. We removed the 14-day smoothing and replotted. Small impacts on statistics were noted as daily noise was reduced previously (e.g. FCH4 0.60 -> 0.53), though qualitatively the same from observing scatter plot distributions.

**Manuscript changes**. Figure 3 was updated

**Comment 4**

"The $R^2$ values indicate good predictive performance of the model and suggest high extrapolation potential. Interpreting why the selected variables in this model effectively explain $CO_2$ and $CH_4$ fluxes would support broader application of the model. Can any explanations be drawn from observational evidence or insights from previous studies?"

**Response**. We agree and have now added a paragraph in the results section to discuss the features found and their relevance to the ecosystem dynamics.

**Manuscript changes**. Inserted paragraph starting 'The feature selection routine converged'

**Comment 5**

"What does the gray grid in Figure 5 represent? It is difficult to understand the spatial extent of the region covered in Figure 5. Additionally, the location and relative distance to the training sites are unclear."

**Response**. Instead of showing the grey area, which was a district of the Delta, we instead overlaid tidal and nontidal waters only. Furthermore, Figure 1 was updated so that areal extents match, so that training sites can be more easily identified.

**Manuscript changes**. Figures 5 and Figure 1 updated.

**Comment 6**

"The manuscript discusses spatial variation based on Figure 5, but it is difficult to interpret this from the figure. Since upscaling is a key component of the study, it is important to clearly present the spatial distribution. While the general north-south differences can be understood, it is hard to assess spatial variation between adjacent areas. It may be helpful to either enlarge specific regions or refine the color scale to better illustrate the variation.

**Response**. We added three zoom-in panels to highlight local variability in high density data regions in the redesigned Figure 5.

**Manuscript changes**. Incorporated into new Figure 5 and updated caption.

**Comment 7** - Prediction validity at extrapolated sites

"I understand the extrapolation capability assessed by cross-validation within the Sacramento Delta. However, in evaluating the validity of the upscaling, I believe it is also important to assess prediction accuracy in areas farther from the three nearby training sites. Even if full time series data are not available, is it possible to evaluate the model's validity using datasets (observations or estimates in previous studies) from different locations or periods?"

**Response**. Observations could not be sourced for locations in the far field for this type of vegetation. Instead, we included a model error term by training 10 models and calculating inter-model mean and standard deviation (see figure S1). Neural network models include stochasticity in the solution search, so each model generates slightly different results.  We used the coefficient of variation (standard deviation / mean) to determine the confidence of the model solution. A higher value shows higher model uncertainty and indicates more unseen conditions.

**Manuscript changes.** Added Fig S1.

**Comment 8** - Causes of early-year discrepancies

"L446-450 'The difficulty in reproducing …' Would it be possible to objectively and quantitatively explain the causes of the discrepancies in the early years using previous studies or available datasets?"

**Response**. Yes. Newly flooded, vegetation-poor wetlands are widely observed to behave as short-term greenhouse-gas (GHG) sources before maturing into sinks. Three long-term studies from the Sacramento–San Joaquin Delta quantify this effect and are cited in the text.

**Manuscript changes**. See line: Restored Delta wetlands are often…

**Comment 9** - Hyper-parameter choices

"Does this framework require any special considerations for determining the models' and hyperparameters?"

**Response**. We extensively searched hyper-parameter spaces for all models and found model defaults were mostly sufficient, with only the SVM method needing slight tuning (reduction of the regularization parameter C, which reduced overfitting).

**Manuscript changes**. Added small paragraph to section 2.4

**Comment 10** - Input-feature resolution

"What are the temporal and spatial resolutions of the input features? How are 4-day MODIS products applied to daily scale predictions?"

**Response**. Methods 2.3 now clarifies: WLDAS is daily at 1 km (no smoothing), Landsat is 30 m nominal 16-day (3×3 pixel average, linear interpolation to daily, 17-day running mean), and MODIS indices were tested but Landsat yielded better skill.

**Manuscript changes**. Added paragraph at the end of 2.3.

**Minor comments:**

| # | Reviewer comment | Action taken |
|---|---|---|
| 1 | "There are some cases where the formatting does not conform to the GMD template …" | We reviewed inconsistencies and updated font mismatches. |
| 2 | "Do the squares on the left plot in Figure 1 exactly match the area shown in Google Earth pictures?" | Verified coordinates; box extents adjusted. |
| 3 | "L382 upscaling.. -> upscaling." | Typo corrected. |

••••••••••••••••••••••••••••••••••••••••••••••••••••••••••••••••

After further investigation, we found that LANDSAT satellite products outperformed MODIS products, scoring higher performance with fewer features for $CO_2$ flux. This improvement is likely due to the much higher spatial resolution (30 m) and the regional scale of the study. The disadvantage is that LANDSAT is available from 2013 (whereas MODIS covers all EC data time periods). Nevertheless, we believe the gains were worth making the change. See comparison below:

**MODIS:**

[Figure]

**LANDSAT:**

---

## Author Response (AR2)

**Comment 1**

"Regarding the response to Comment 4:

The addition of explanations to the Results section is good. In addition, it would be beneficial to discuss in the Discussion section why the properties of these features were able to explain FCO2 and FCH4 well. It might also be helpful to include ecological or geoscientific reasoning, for example by referring to the specific characteristics of the Sacramento Delta.

This could help address potential concerns from readers about how such a small number of features (only two or three) could result in high predictive accuracy. Note that this comment is not a request for complete proof.

In addition to representing water conditions, is NDGI also related to the amount of vegetation? If so, could this have contributed to its importance in explaining FCH4?"

**Response:**

We agree and the discussion section was expanded to highlight the mechanistic relevance of the predictors for both targets, as well as including more information relating to the relevance of NDGI on methane, which is indeed important since substrate is supplied from the vegetation, which promotes methanogenesis.

**Manuscript changes.**

• We added a paragraph to expand the discussion section, which highlights the features selected. See 'The feed-forward selection converges...'

**Comment 2**

"Figure 5: I believe the figure has been improved. However, FCH4 has been removed. Why is that?"

**Response.**

This was a stylistic choice as Figure 5 became busy with the updated zoom-in panels, and FCH4's contribution is contained within FCO2e.

**Comment 3**

"... zoom-in subplots highlighting areas with more data. 10 models were trained and mean and standard deviation was calculated for each spatial point."

This part feels abrupt and lacks a smooth connection between sentences. Also, what do the ten models refer to? I was unable to understand it."

**Response.**

Fair point, the current sentence is ambiguous, and I made this clearer. To clarify, we run N separate machine models on the same data (there is stochasticity in the training), then N separate upscalings were calculated from each of the models, and the ensemble mean is reported.

**Manuscript changes.**

• Updated Section 3.2 to make this clearer. See 'Upscaling was repeated...'

**Comment 4**

"Regarding the change to LANDSAT

The revised model now seems to select features that offer more reasonable explanations of the phenomena in wetlands. Therefore, I agree with this change."

**Response.**

Great!